# Effects of Environmental and Socioeconomic Inequalities on Health Outcomes: A Multi-Region Time-Series Study

**DOI:** 10.3390/ijerph192416521

**Published:** 2022-12-09

**Authors:** Iara da Silva, Caroline Fernanda Hei Wikuats, Elizabeth Mie Hashimoto, Leila Droprinchinski Martins

**Affiliations:** 1Graduate Program in Environmental Engineering, Campus Londrina, Federal University of Technology—Paraná, Av. Dos Pioneiros, 3131, Londrina 86036-370, Paraná, Brazil; 2Department of Atmospheric Sciences, Institute of Astronomy, Geophysics and Atmospheric Sciences, University of São Paulo, Rua do Matão, 1226, São Paulo 05508-090, São Paulo, Brazil

**Keywords:** temperature extremes, socioeconomic conditions, climate and health, regression analysis, relative risk

## Abstract

**Highlights:**

**What are the main findings?**
The maximum temperature and relative humidity are the variables of greatest risk for cardiovascular diseases.The minimum temperature represents the highest risk variable for respiratory diseases.Mental diseases are influenced by extreme temperatures.The relative risk varied among regions as a function of the socioeconomic conditions and climate.

**Abstract:**

The gradual increase in temperatures and changes in relative humidity, added to the aging and socioeconomic conditions of the population, may represent problems for public health, given that future projections predict even more noticeable changes in the climate and the age pyramid, which require analyses at an appropriate spatial scale. To our knowledge, an analysis of the synergic effects of several climatic and socioeconomic conditions on hospital admissions and deaths by cardiorespiratory and mental disorders has not yet been performed in Brazil. Statistical analyses were performed using public time series (1996–2015) of daily health and meteorological data from 16 metropolitan regions (in a subtropical climate zone in South America). Health data were stratified into six groups according to gender and age ranges (40–59; 60–79; and ≥80 years old) for each region. For the regression analysis, two distributions (Poisson and binomial negative) were tested with and without zero adjustments for the complete series and percentiles. Finally, the relative risks were calculated, and the effects based on exposure–response curves were evaluated and compared among regions. The negative binomial distribution fit the data best. High temperatures and low relative humidity were the most relevant risk factors for hospitalizations for cardiovascular diseases (lag = 0), while minimum temperatures were important for respiratory diseases (lag = 2 or 3 days). Temperature extremes, both high and low, were the most important risk factors for mental illnesses at lag 0. Groups with people over 60 years old presented higher risks for cardiovascular and respiratory diseases, while this was observed for the adult group (40-59 years old) in relation to mental disorders. In general, no major differences were found in the results between men and women. However, regions with higher urbanization levels presented risks, mainly for respiratory diseases, while the same was observed for cardiovascular diseases for regions with lower levels of urbanization. The Municipal Human Development Index is an important factor for the occurrence of diseases and deaths for all regions, depending on the evaluated group, representing high risks for health outcomes (the value for hospitalization for cardiovascular diseases was 1.6713 for the female adult group in the metropolitan region Palmas, and the value for hospitalization for respiratory diseases was 1.7274 for the female adult group in the metropolitan region Campo Mourão). In general, less developed regions have less access to adequate health care and better living conditions.

## 1. Introduction

Climate change is the result of natural or man-made actions that can be harmful to the planet, such as the accumulation of excess greenhouse gases in the atmosphere by various human activities or the improper use of soil, water, and natural resources. Therefore, it can be defined as the manifestation of the human impact on the ecosystem [1,2]. Climate change has been presented as one of the most challenging problems in the world’s 21st-century scenario and is directly linked to public health problems [3,4,5,6].

Human beings interact with the environment in which they live, influencing and altering it, besides receiving influences. Sudden variations stimulate responses in the body [7,8], which can frequently represent damage to an individual’s health, depending on the intensity of the stimulus [9,10]. The body adapts to the frequent climatic conditions of the region; nonetheless, in the face of anomalous phenomena, the most sensitive individuals tend to have higher disease rates related to environmental perception variables [11].

Air temperature and relative humidity, for example, are climatological variables directly linked to thermal sensation [12,13]. Thus, diseases related to the perception of heat and cold will have their occurrence index influenced by the fluctuations in these variables, such as those linked to the circulatory and respiratory systems [14,15,16]. The relationship of temperature with cardiovascular and respiratory diseases, which leads to an excess of deaths and hospitalizations, and its influence more specifically in certain age groups has been widely explored in epidemiological studies worldwide [15,17,18,19,20]. Human adaptability to the local climate is also an important factor in the climate–mortality ratio [21]. In addition, there are studies relating mental disorders to thermal extremes as a function of the direct influence of temperature on the neurological system [22,23,24].

Changes in the environment are felt by the entire population. However, there are groups that are more affected by these variations due to individual and age characteristics, such as individuals with physiological weaknesses, e.g., children, the elderly, and the chronically ill [25,26]. A decrease in metabolic activity with age alters the speed and intensity of the reaction stimulated by external variables [27,28]. The human body responds to heat stress by limiting elevations in core temperature. The exacerbation of mechanism activation to control temperature can lead to several outcomes, mainly in those with physiological weaknesses [6].

The congruence between population aging and increased extreme weather events represents higher disease rates directly or indirectly linked to climate change. This scenario leads to higher expenses for the state for treatments and hospitalizations that present a diagnosis of these diseases. Furthermore, socioeconomics make low-income people a more vulnerable population to climate change since they generally do not have clinical follow-up, diagnosis, or early treatment of diseases [29]. The COVID-19 pandemic exposed these inequalities and evidenced the incapacity of the world to deal with them [30]. In addition, social standardization is observed in hospital admissions, especially for mental and behavioral illnesses that suffer from social stigmatization [31].

Population aging is a trend observed worldwide, with a projection that it will double its proportion in 2050 compared to 2015. However, the highest rates are in low- and middle-income countries [32], such as in South America and Africa. For example, the Brazilian elderly population in 2000 represented 8.2% of the total; it will be 18.6% in 2030 [33]. Likewise, cardiovascular diseases cause approximately 18 million deaths per year worldwide, of which about 80% are related to low-income people. Thus, studies on these relationships are essential on an appropriate spatial scale that makes it possible to determine more assertively the associations and risks between environmental and health variables, considering the climate, the populations, and the socioeconomic conditions of each region [31,32]. Therefore, the objective of this study was to evaluate, with a high resolution, the long- and short-term effects of temperature (maximum and minimum), relative humidity, and socioeconomic conditions on hospitalizations and deaths due to cardiovascular, respiratory, and mental diseases in adults and the elderly of both genders. Nineteen years and 16 subregions were analyzed, highlighting the regional variability in the associations for a subtropical climate region.

## 2. Materials and Methods

### 2.1. Study Area

The area of study is the Paraná state, which presents different climates, levels of economic development, and land use. Figure 1 shows the area, which is 199,880 km^2^ and has more than 11 million inhabitants [34]. The main economic activities are agriculture (soy, coffee, corn, sugarcane, and wheat), industry (agribusiness, automobile, paper, and cellulose), and plant extraction (wood and yerba mate) [34,35]. It is one of the principal regions of South America for the production of food. There are three types of climates: subtropical Cfa on the coast, with well-distributed rainfall throughout the year (≅1500 mm/year) and high temperatures in the summer; in the highest areas of the state, there is regular rainfall (≅1200 mm/year) but mild summers, characterizing the Cfb subtropical climate; and in the northwest, summers are intense with high temperatures and concentrated rains, while winters tend to be dry [36,37,38].

The area was divided into 16 metropolitan regions (MR), of which eight are official (Figure 1): Curitiba (2, MRC), Londrina (5, MRL), Maringá (6, MRM), Apucarana (7, MRA), Campo Mourão (8, MRCM), Umuarama (9, MRU), Toledo (10, MRT), and Cascavel (11, MRCA). The other eight MRs were proposed, and as for the official regions, were named after the main cities: Paranaguá (1, MRPR), Telêmaco Borba (3, MRTB), Bandeirantes (4, MRB), Foz do Iguaçu (12, MRF), Francisco Beltrão (13, MRFB), Pato Branco (14, MRPB), Palmas (15, MRP), and Guarapuava (16, MRG). This proposed classification was carried out according to the geographical, population, socioeconomic, and climatic characteristics of the cities that compose the MR, representing 84% of the Paraná state area and 85% of its population [39].

### 2.2. Data Collection

The meteorological data used in the study came from IAPAR stations (Instituto Agronômico do Paraná—Paraná Agronomic Institute), SIMEPAR (Sistema Meteorológico do Paraná—Meteorological System of Paraná), and INMET (Instituto Nacional de Meteorologia—National Institute of Meteorology). The analyzed variables were daily averages of maximum and minimum temperatures (°C) and relative humidity (%), which are directly related to thermal sensation [40]. There was at least one meteorological station for each metropolitan region that was studied, totaling 23 stations used in the study.

The health data consisted of the daily number of hospital admissions (the public health system and the private systems affiliated to the SUS—Unified Health System) and deaths (public and private health systems) of adults and the elderly registered between 1996 and 2015 in the regions under study. The hospitalization data are freely available on the online platform of DATASUS, the Department of Informatics of the SUS, and the death data are available on the online platform SIM, the mortality information system of the SUS.

The studied cases presented diagnoses of diseases belonging to three chapters of the international code of diseases (ICD10): chapter IX of cardiovascular diseases (I00-I99); chapter X of respiratory diseases (J00-J99); and chapter V of mental illnesses and disorders. The numbers of daily cases of hospitalizations and deaths were stratified into three age groups (40 to 59 years old (1), 60 to 79 years old (2), and over 80 years old (3)) and by sex and were thus compiled into six groups that are henceforth referred to as GM1, GM2, and GM3 for men (i.e., GM means group of men) and GW1, GW2, and GW3 for women (i.e., GW means group of women) according to the mentioned age groups that were designed from 1 to 3. This division was based on the biological characteristics of the organism that determine physiological reactions [26,41,42]. The average data period was 19 years (1996–2015), with four regions (MRA, MRG, MRT, and MRU) with data series until 2012 and the MRF with data from 1999.

Socioeconomic information was also analyzed. For this, municipal human development data were collected for all municipalities that were studied [43]. They used the Municipal Human Development Index (MHDI), which considers longevity (life expectancy), education (average years of study), and income (average per capita income in the municipality), ranging between 0 (no development) and 1 (total development). The index was calculated in specific years for each municipality and was made available by the Paraná Institute for Economic and Social Development—Instituto Paranaense de Desenvolvimento Econômico e Social—IPARDES [44]. The methodology to calculate the index is laborious and follows a similar line to the Human Development Index (HDI) prepared by the United Nations Development Programme (UNDP). A brief of several steps is presented in the Appendix A. For inclusion in the models, these municipal indices were used to calculate an average value for the metropolitan region from the specific information of each municipality.

### 2.3. Statistical Methods

A time-series study was performed based on a regression analysis with the generalized additive model for location, scale, and shape (GAMLSS) combined with an exponential probability distribution. Poisson and negative binomial distributions were tested, as they are widely used in this type of study, and the zero-inflated model was also tested. The distribution that best fit the dataset was selected using the Akaike information criterion—AIC [45]. This criterion is a metric that measures the quality of a statistical model, through which it is possible to compare the tested distributions, and lower AIC values represent a higher quality of fit for the final model considering the distribution and the independent variables (see Appendix A). The selection of variables to compose the final model was carried out using the stepwise method, in which, at each stage, all variables in the model were previously verified by their partial statistics and were inserted or removed according to their significance in the sample [46]. The independent variables were selected because they had different statistical significance as we analyzed temperatures and humidity in different regions and groups. The variables selected for each group and the AIC for the distributions are presented in the Appendix A.

The multicollinearity of the selected variables is a problem for estimating the regression coefficients [47]. To solve it, the GVIF (generalized variance inflation factor) proposed by [48] was observed. GVIF values between 5 and 10 indicate that the variables are highly correlated. In these cases, the principal components technique was applied. This method uses an orthogonal transformation to join correlated variables into a new set of uncorrelated variables [49,50].

In the models for hospital admissions, two variables were added: *holiday* (with 0 for non-holidays and 1 for national holidays) and *weekday* (starting with 1 for Sunday, 2 for Monday, and so on)—to control the trend of hospitalizations on working days observed in other studies [51,52,53] and the analyzed database. These variables were not added to the mortality models, as this behavioral effect does not impact the number of deaths. Besides meteorological variables, a socioeconomic variable was inserted in the model to compare socioeconomic conditions’ influence on MR outcomes. Meteorological variables show seasonality, which are periodicities specific to the seasons. Therefore, a penalized spline was used to smooth this particularity in all models.

For the systematic part, the adjusted semi-parametric model is given by:$$g\left(\mu_{i}\right)=\alpha +\beta_{1}x_{1i}+\beta_{2}x_{2i}+\beta_{3}x_{3i}+\beta_{4}x_{4i}+\beta_{5}x_{5i}+\beta_{6}x_{6i}+h\left(time_{i}\right),\;i=1,\ldots ,\;n,$$
where *g*(.) is a logarithmic link function; α is the intercept; $\beta_{j}\left(j=1,\ldots ,6\right)$ represents the *j*th regression coefficients; the meteorological and socioeconomic explanatory variables are: *x*_1_—the maximum temperature, *x*_2_—the minimum temperature, *x*_3_—the relative humidity, *x*_4_—the holiday, *x*_5_—the weekday, and *x*_6_—the MHDI; time is from 1 to *n* days in the series; and *h*(.) represents the penalized splines. The associations of temperature and relative humidity were also investigated in the plots for data of the 25th, 75th, and 99th percentiles and above the 99th percentiles for all groups. The relative risks (RR) were calculated from the β coefficients of the final adjusted models for each variable. After defining the final model, the socioeconomic variable MHDI was analyzed as a predictor variable, and the RR at a 95% confidence interval (CI) was calculated for the proposed groups in each region.

The short-term exposure was also assessed, as the effects of exposure to meteorological variables are not linear and are not instantaneous and may take several days to affect health [54], especially in the case of temperature, which was observed in several studies [20,23,24,55]. Up to seven days of lag were considered for all groups, regions, and ICDs of studied diseases, and risk curves were plotted using the distributed lag linear and non-linear models (DLNM) package [56]. DLNM has a modeling framework that represents exposure–response and time-lag effects simultaneously. Combined with GAMLSS, it provides values of an event with a lag of *n* days, and the cumulative exposure effect over the period [16,57]. In summary, the distributions and predictive variables were determined for the final model, from which the percentiles and risk curves with time lag were analyzed for all groups. A flowchart of the performed steps is presented in Figure 2. The statistical analyses were performed using the software R, version 3.6.1.

## 3. Results

### 3.1. Description and Characteristics of the Study Variables

The climate characterization graphs, the health indices, and the tables with the RR, AIC values, and the variables selected by the stepwise method are presented in the Appendix A.

For the study period, the coldest regions were in the south and central part of the area, with the MRC and MRP (south part) being the coldest. The hottest regions were observed in the state’s north, northeast, and west, with the MRT and MRFB having the highest temperatures and the greatest thermal amplitude. Concerning relative humidity, the coastal regions, part of the northeast, and the MRF were the most humid, mainly due to the water bodies present in the vicinity of these areas (Appendix A). MRPR was the most humid and had the lowest relative humidity variation throughout the year. The driest portions of the state coincided with the highest temperatures.

The MRA, MRB, MRCA, MRCM, and MRU had higher incidences (number per 100,000 inhabitants) of hospitalizations for diseases of the cardiovascular system. The MRFB, MRF, MRPR, and MRC has the lowest rates (Appendix A), which was repeated for the hospital admissions for respiratory diseases in the MRPR and MRC regions (Appendix A). In contrast, the MRC had the highest incidence of hospitalizations for mental illnesses and disorders, followed by the MRCM, MRPR, and MRTB (Appendix A). The MRA, MRB, MRCA, MRCM, and MRU also had higher incidences of deaths from cardiovascular problems (Appendix A), while deaths from respiratory diseases had greater variability, with an average value for the studied regions of 70 deaths for every 100,000 inhabitants (Appendix A).

The socioeconomic index calculated based on income, employment, production, health, and education provided by IPARDES, as mentioned since 2010, has been indicative of the development of municipalities in the state and was divided into four categories: 0.00 to 0.39 (low performance); 0.40 to 0.59 (medium-low performance); 0.60 to 0.79 (mean performance); and 0.8 to 1.0 (high performance). The average values of the index (Appendix A) made it clear that the state’s health conditions are generally in the medium to high-performance category; on the other hand, income remains mostly in the low-performance class. Educational development was the most variable social dimension between cities and years of data collection [34]. The analyzed regions showed variability in the development index, with more and less developed cities in the same region.

The Paraná state is highly urbanized, with approximately 85% of the population living in urban areas. Most cities have an urbanization rate above 55%; however, some cities have a lower rate. The urbanization rate tends to be higher in cities with higher income generation; therefore, more developed areas were concentrated in the north and west of the state and some specific cities such as the capital of Curitiba and the coastal city of Paranaguá [34].

### 3.2. Regional Variability in Short- and Long-Term Effects of the Environmental and Socioeconomic Conditions on Health

For most of the MRs, the negative binomial distribution best fit the dataset (56.2% of models) based on the AIC values (Appendix A), followed by the Poisson distribution (40.5%). Only the groups with excess non-recording of the outcome (>30% null values) showed a better fit when using distributions with zero inflation (1.7%). The lack of fit could lead to misinterpretations of results and therefore should be previously evaluated.

Only four models among the studied regions presented problems with multicollinearity: MRG’s GM2—hospitalizations for cardiovascular diseases; MRG’s GM2—hospitalizations due to respiratory diseases; MRU’s GM2—deaths from cardiovascular diseases; and MRU’s GM2—deaths from respiratory diseases.

The highest record of hospitalizations on weekdays is a bias that was observed in this study and was also found in other studies [51,52,53]. It is associated with the type of treatment and the need for more intensive care [58,59]. This bias was not observed for deaths.

#### 3.2.1. Cardiovascular Diseases

Hospital admissions related to cardiovascular diseases were directly associated with the maximum temperature and relative humidity (Appendix A), with an increase of up to four times in the risk of occurrence of the health outcome associated with the meteorological variable, as observed for the GW1 group of MRL at lag 1 (1.0410, CI: 1.0032–1.0803), especially in the north and west parts of the study region, which were the warmest and driest areas of the state. In contrast, deaths related to these diseases were more impacted by the minimum temperatures (Appendix A) since this variable was selected more frequently to compose the final model, with almost three times increased risk in MRA’s GW1 group for the minimum temperature at lag 0 (1.0276, CI: 1.0028–1.0530). Regarding hospitalization, considering all regions, there was a higher percentage of risk factors (meteorological and socioeconomic explanatory variables) for the groups GW2 (68.8%), GM2 (62.5%), and GW3 (62.5%) (Appendix A). However, for deaths, there was a higher risk fraction for the younger group, including both men (GM1) and women (GW1), representing 43.8% and 31.3%, respectively (Appendix A). The RR curves with the respective 95% CIs for the main lags and the studied groups showed that the risk was mainly associated with higher temperatures and lower relative humidity (Figure 3), which represent the riskiest environmental conditions for the occurrence of cardiovascular problems. This was also observed in the final regression models adjusted for the percentiles (25th, 75th, and 99th) for the regions, as seen in Figure 4. Figure 3 shows the relative risks of hospitalizations and deaths associated with relative humidity and maximum and minimum temperatures at lag 0 for some regions and groups. Temperatures above the median of the series (reference value) presented a risk for the occurrence of cardiovascular diseases, while for humidity values below the reference presented a risk.

When the final regression models were adjusted for the percentile ranges, the results were similar to those shown by the risk curves (Figure 3). For temperatures, the 75th and 99th percentile ranges were at risk, especially in the last range consisting of the upper extremes of temperature, while for humidity the 25th percentile was more significant (Figure 4). The combination of higher temperatures and low relative humidity represented the riskiest environmental conditions for the occurrence of cardiovascular problems.

Concerning the socioeconomic characteristics evaluated in the MHDI, the risks of hospitalization were more expressive in regions with less developed municipalities, such as MRA (GW2 group—1.3611, CI: 1.2733–1.4549) and MRTB (GW3 group—1.1319, CI: 1.0250–1.2499), and presented a protection factor in regions with higher development indices, such as MRCM (GW1 group—0.7948, CI: 0.7455–0.8474) and MRC (GW3—0.7285, CI: 0.6653–0.7977).

#### 3.2.2. Respiratory Diseases

Hospitalizations and deaths from diseases of the respiratory system were directly related to the minimum temperature for the entire state, which was a more frequently significant variable in the studied groups (Appendix A). However, risks for hospital admissions were also related to the maximum temperature. For the MRB’s GM3 group, the increase in risk was 2.7 times greater for this meteorological variable (1.0269, CI: 1.0103–1.0438). As for deaths, the minimum temperatures were found to be a significant variable for the final model (Appendix A), with the risk of occurrence increasing almost four times in some regions (MRF’s GM1 group at lag 3—1.0370, CI: 1.0137–1.0610) and the selected humidity presenting a risk (MRB’s GW1 group at lag 0—1.0222, CI: 1.0074–1.0373). The RR for diseases in this chapter was higher for groups over 80 years (GM3 and GW3). Women in this age group had 68.8% of risk factors in the regions, while for men it was 50%. For deaths and respiratory diseases, the groups of 40- to 60-year-olds were at constant risk from environmental and socioeconomic variables, with 43.8% of the risk factors, considering all regions for GM1 and 31.3% for GW1.

The risk curves for respiratory diseases (Figure 5) confirmed the incisive effect of the minimum temperatures, especially in the hottest regions of the state, such as the region of Maringá, while a risk even closer to the temperature of thermal comfort was observed in deaths. Therefore, the risk was noticed in the 25th percentile range of the minimum of these regions (Figure 6). On the other hand, the maximum values had the risk concentrated in the right part of the curve, close to the highest recorded values. Respiratory diseases at low temperatures or extremely high temperatures with low humidity are caused by stress in the respiratory system [60,61,62].

Socioeconomic characteristics represented a direct impact on respiratory diseases, especially in the more developed regions of the state, such as MRB (GW1 group—1.1849, CI: 1.0195–1.3774), MRCM (GW1 group—1.7274, CI: 1.5726–1.8974), MRCA (GW3 group—1.0704, CI: 1.0395–1.2049), and MRPR (GM1 group—1.2769, CI: 1.1579–1.4082), with groups of women appearing to be more susceptible than men.

#### 3.2.3. Mental Illnesses and Disorders

Hospitalizations for mental illnesses and disorders were not recorded for all metropolitan regions. As mentioned, the MRC and MRCM presented the highest occurrence rates of these hospital admissions. The groups of men and women aged 40 to 60 years were at higher risk for variables investigated among the regions that registered cases of these diseases (Appendix A), with 63.7% for both sexes. Instantaneous effects were more common, but risks of up to five days of lag were frequent.

The maximum temperature represented an increase of more than 16 times in the risk of occurrence of mental illnesses and disorders for the MRCA’s GW2 group at lag 5 (1.1651, CI: 1.0607–1.2797), while for the minimum temperature the increase was approximately 12 times for the MRC’s GW2 group at lag 0 (1.1194, CI: 1.0892–1.1505). The relative humidity increased almost three times for the MRF’s GW1 group at lag 0 (1.0253, CI: 1.0115–1.0394).

The exposure–response curves (Figure 7) made it clear that even at minimum temperatures, the greatest risk was observed at the highest values, and the 75th and 99th percentile ranges showed risk more frequently (Figure 8). Although most regions followed this behavior, some diverged from this pattern, such as MRPR and MRCM for the GM1 and GW1 groups, respectively.

It is difficult to measure the impact of socioeconomic status on mental illnesses, as access to such care is already restricted to groups and areas of greater development, in addition to the stigmatization of this type of disease [63]. The records show that young people sought more follow-up and treatment for mental illness and disorders than older people and women. For the studied regions, MRC presented the MHDI as a protective factor for most groups.

## 4. Discussion

The maximum and minimum temperatures showed a tenuous and constant increasing tendency in most regions, and the relative humidity varied according to the region and the rainfall rates [64,65,66]. The climate of the study area, as well as that of Brazil, showed an increase in the thermal amplitude, making it possible to observe regional differences in climate and health outcomes, which was also observed by [67].

Monthly hospitalization averages for cardiovascular diseases were higher in the summer months (November to February); for respiratory diseases, the data peaked in the cold months, mainly in June, which is a period of transition from fall to winter (winter officially starts on June 21), which was also observed in other studies [68,69]. Mental illnesses and disorders had irregular hospitalization rates throughout the year, but the summer and spring months had the highest rates, as also observed in other studies [70,71]. High temperatures were associated with an increased risk for hospital admissions for cardiovascular diseases, while minimum temperatures were more related to hospital admissions for respiratory problems. Similar results were also found by [5] for rural villages in Northwestern China.

The most developed MRs were concentrated in the north and west of the state, such as the MRM, MRF, and MRL, which had the highest MHDI values. The Palmas, Guarapuava, and Telêmaco Broba regions had the lowest rates. Despite this classification, MHDI values are averages, with more and less developed cities in the same regions. Despite having a high MHDI, Curitiba, the state capital, had poorly developed cities in its metropolitan region, which was reflected in the average of the MR index. The most developed regions presented the highest risk for respiratory problems, except Maringá, while less developed areas pointed to more frequent risks for cardiovascular diseases. Observing the climatic variables, this pattern was confirmed when calculating the RR for the socioeconomic variable, with different impacts on the analyzed groups.

The study of the effects of socioeconomic conditions on hospitalizations and deaths from cardiorespiratory and mental disorders, respecting the diversity of regions with different climatic conditions in Brazil, was interesting, given the plurality of the country. This socioeconomic effect was also observed in studies carried out on other continents [19,72,73]. Regions with higher development indices generally present a protection factor for cardiovascular diseases, a result corroborated by other studies in other countries in which socioeconomic differences represent a difference in the occurrence of cardiovascular diseases according to social class [63,64]. As for deaths caused by cardiovascular diseases, the MHDI had less impact on health outcomes. In general, socioeconomic inequality had a greater impact on women, a result similar to that found in a study in England [74].

At the same time, as the areas with the highest MHDI were also the most urbanized, the direct relationship of this factor with airway diseases was probably related to the air quality of these cities. In [75], the authors observed the effect of the socioeconomic level on the occurrence of respiratory diseases and observed that differences in urban development impact air quality and consequently occurrences, corroborating the results obtained in this study, except for the greater impact observed in men. Therefore, the socioeconomic position can function as an independent determinant of the health of the respiratory system for the regions of the state, as observed in other studies [61,66]. In addition, in [17] the authors reported that the poverty rate and low socioeconomic development increased the risk of hospitalization due to high temperatures. Meanwhile, higher household incomes and access to drinking water showed reduced risks related to climatic variables.

The minimum temperature was a determining factor for deaths in different age groups concerning cardiovascular diseases, for both men and women, as also found by other authors [76,77]. Despite this characteristic, in the hottest areas of the state, the maximum values of temperature and lower relative humidity presented greater risks. This finding highlighted that the intensity of temperature and humidity varies among regions since, even though the organism has certain adaptability, the climate has been warming relatively fast due to global and local climate changes. Furthermore, mitigation measurements for temperature, especially for low- to medium-income people, are non-existent. Therefore, the socioeconomic conditions and climate justify these differences among regions.

Although it takes time for the organism to assimilate the stimuli of the environment, lag zero is the one that presents the most frequent risk for cardiovascular diseases, especially for temperature. In their study in Brazil, [62] also observed a greater effect of temperature at lags zero and one for hospital admissions. The lags related to heat are smaller than those related to cold for the elderly, with the delay for the maximum temperature varying from zero to one day, while for the minimum this variation is commonly shifted by two to three days [15]. When the body is exposed to very low temperatures, there is an increase in catecholamines (neurotransmitters that initiate the process of vasoconstriction and tachycardia), which increases blood pressure [78,79,80]. In the long run, these biological effects can result in increased blood viscosity and myocardial ischemia [81].

Exposure to heat, on the other hand, causes vasodilation as a response mechanism to reduce the increase in body temperature, which demands greater effort from the cardiovascular system, raising the heart rate and reducing the blood volume in the atria, chest (heart, thorax, and veins), liver, and spleen. This blood deficit causes heart failure, which, combined with thermal stress, induces increases in erythrocytes, neutrophils, and platelets, increasing blood viscosity and damaging the cardiovascular system [29,81].

In addition to the instantaneous effect, cardiovascular diseases, present significant RR from 48 to 96 h, as found in other studies [73,82]. Hospitalizations for this disease chapter occurred more frequently at lag 0, while deaths showed high frequency in lags after two days, a result also observed by [15]. This difference found between hospitalization and death may be related to treatments and clinical interventions and the time necessary for thermal stress to be fatal to the body [83].

The higher temperatures (75th and 99th percentiles) presented a risk, in contrast to relative humidity, which proved to be a risk factor in the range with lower values (25th percentile), further corroborating the results found in other studies around the world [20,84,85,86,87,88]. Temperature variability over short periods has adverse health effects, which can be enhanced by relative humidity, as it modifies the temperature relationship with the organism [69,87,89].

In [84], the authors indicated physiological adaptation as a response of the population’s acclimatization to changes in temperature (intrinsic adaptation) or to non-climatic factors that contribute to risk reduction (extrinsic adaptation), such as socioeconomic development or the improvement of health services. Therefore, the differences in climate and development between the regions of the state impact the analyzed health outcomes. Although the state has a predominantly humid subtropical climate, the coldest areas are concentrated in the southern portion of the interior plateaus. Therefore, the populations residing in these areas are more fragile to the extremes of maximum temperature, such as in the regions of Paranaguá, Palmas, and Francisco Beltrão.

The groups over 80 years old presented the highest risk for hospitalization due to cardiovascular diseases: women with 68.8% and men with 50%. According to [90] and [83], thermoregulation is not as efficient in elderly individuals, increasing the stress caused by heat and cold. Exposure to the environment causes physiological changes, such as hot conditions that can cause cardiac and thermoregulatory deficits, mainly in the geriatric population, given the physiological and metabolic changes naturally suffered due to aging [91]. However, the younger groups had significant values for deaths from cardiovascular diseases (43.8%—GM1 and 31.3%—GW1). Lethality in groups of 40- to 60-year-olds is conditioned by the heart’s pumping capacity, which supplies the necessary blood flow to the body and gradually decreases with age. Younger people have a higher heart rate; therefore, the occurrence of any problem that alters the blood supply to the heart presents acute results and, associated with some factors such as alcohol consumption, smoking, stress, and physical inactivity, can be lethal [92,93,94,95].

For respiratory diseases, as mentioned, hospitalizations and deaths were directly related to the minimum temperature for the regions. In colder areas, the effect of hot days was greater than that of cold ones, which differs from some studies carried out in Europe, North America, and Asia, in which the effect of high temperatures was more harmful for the occurrence of respiratory diseases [96,97]. For mortality, the extremes of both cold and heat presented risks, a result also observed by [15] in countries in North America, Europe, and Asia. In [98], the authors also found mortality risks associated with exposure to extreme heat and cold temperatures in Brazilian cities. The increase in deaths in winter is due to physiological changes that alter cellular and hormonal immunity, in addition to behavioral factors [90]. As noted, minimum temperatures represented a greater risk for deaths, especially in the north and northwest regions, considering the state’s climate and the adaptation of individuals, which is a consistent response [68].

The RR associated with longer lag intervals for respiratory diseases corroborated those observed, for example, by [15]. Low relative humidity is a risk factor for respiratory diseases, mainly due to the relationship between humidity, air pollutants, and the change caused by humidity in stimulating body temperature [69]. The combination of low humidity and temperature extremes, both low and high, impacts the performance of the respiratory system [99]. It is also necessary to consider outbreaks of influenza viruses related to low humidity, since inhaling dry air impairs the repair of connective and muscular tissue in the system and inhibits mucociliary clearance [100,101].

When the individual is exposed to extreme environmental variables, several biological mechanisms are initiated to stabilize the body temperature around 37 °C, a process called thermoregulation [102,103,104]. Prolonged exposure to these variables affects brain functions. In the brain, heat is produced when oxygen is consumed, and it is removed by blood flow. However, changes in temperature and blood viscosity alter the brain’s cooling process, reducing its efficiency and resulting in damage and changes in the organ [105,106].

In the study area, mental illnesses and disorders had higher hospitalization numbers in the MRC, which is a cold region, and the minimum temperatures presented greater risks for the occurrence of mental disorders, while some studies with different climate conditions reported associations between high temperatures and mental disorders [23,107,108]. In [109], the authors pointed out that, in Brazil, mental disorders were becoming increasingly common in hot regions, highlighting the influence of urbanization and the development of civilization on these diseases. Although the MRC is urbanized, it is one of the coldest in the state. In [110], the authors deeply studied the associations of mental illnesses in Curitiba and found that temperature extremes (both high and low) are considerable risk factors for mental disorders and diseases, with the existence of a dynamic relationship between pollutants and temperature. Similarly, one study carried out in three cities with a subtropical climate in China found that low temperatures had a significant and prolonged effect on mental disorders, mainly schizophrenia, anxiety, and depression [111].

In the Campo Mourão region, maximum temperatures presented greater risks for mental illness. Although it is in a warmer area, the average highs in the region can reach 30 °C. Therefore, ambient temperature exceeds the thermal comfort temperature, causing malaise and affecting mental health, which is aggravated by global warming and the greater intensity and frequency of heatwaves. Temperature affects mental health differently than physical health, as thermal discomfort can cause sleep problems and affect brain activity [112]. Other regions also presented risks, which were quite variable in relation to the maximum and minimum temperatures, indicating that both are risk factors.

Groups aged between 40 and 60 years old had the highest percentages, in both genders, for mental illness (63.7%). This age group is identified as the most affected in other countries, which can be extended to 75 years [20,22,106,113]. It is necessary to take into account the social and cultural stigmatization of mental illnesses and disorders, which are frequently not properly diagnosed or treated [63,114].

The influence of temperatures on brain structures is highly variable. For most regions, lag zero was more significant regardless of climate and socioeconomic conditions, but risks of up to five days were still observed. In [82], the authors also found significant associations for the period from zero to four days, with risks still being observed until a delay of seven days from exposure, while [22] concluded that the short-term effect of temperature associated with mental illness is varied and depends on the diagnosis.

## 5. Conclusions

The climate of the study region (Paraná state) tends to experience changes, such as an increase in the current temperature, which is estimated in future climate projections. Older age groups were more sensitive to environmental exposure, representing the largest share of cardiovascular and respiratory patients. As for mental illnesses and disorders, younger groups were the ones with the highest incidence. No statistically significant differences were found in the registered number of cases between men and women; however, women had a higher percentage of groups with a RR for cardiovascular and respiratory diseases.

Hospital admissions for cardiovascular diseases were more impacted by the combination of high temperatures and low relative humidity. On the other hand, hospitalizations for respiratory diseases are associated with both upper and lower temperature extremes and low relative humidity. Mental disorders are associated with extremes of both minimum and maximum temperatures.

The MHDI value is a factor that mainly affects the hospitalization rate for cardiovascular diseases in the less developed areas of the state. Conversely, the areas with the highest development rates presented the greatest risks for respiratory diseases. This divergence was probably related to the low air quality in the most urbanized regions (Figure 9).

Paraná is going through a heating process. The northern and northwestern regions of the state, naturally warmer and more urbanized, are warming more sharply than the southern regions. Although the populations in the hottest places are adapted to the higher temperatures, the rapid increase can cause discomfort and health problems. On the other hand, in the coldest regions of the state, the increase in minimum temperatures causes thermal stress, especially in the most physiologically fragile people, a number that may also increase in the coming years with population aging, which is an observed trend. The warming and the alteration of the state’s rainfall from climate change and land use are directly linked to air quality, along with other factors, such as urbanization. This environmental condition, combined with the increase in the most fragile part of the population, represents higher rates of diseases linked to the environment and, consequently, increased spending for the state’s health system and the loss of human lives.

## Figures and Tables

**Figure 1 ijerph-19-16521-f001:**
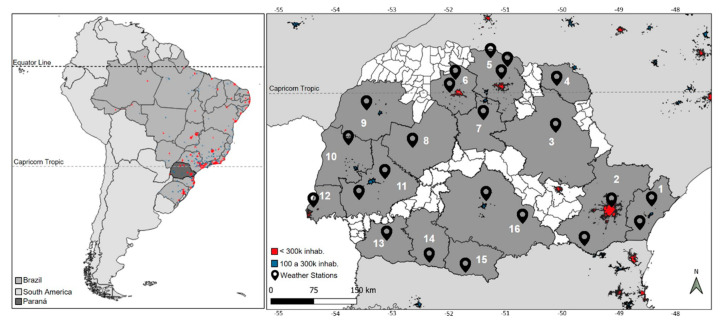
Study area and metropolitan regions used in this study. MRs: Paranaguá (1), Curitiba (2), Telêmaco Borba (3), Bandeirantes (4), Londrina (5), Maringá (6), Apucarana (7), Campo Mourão (8), Umuarama (9), Toledo (10), Cascavel (11), Foz do Iguaçu (12), Francisco Beltrão (13), Pato Branco (14), Palmas (15), and Guarapuava (16).

**Figure 2 ijerph-19-16521-f002:**
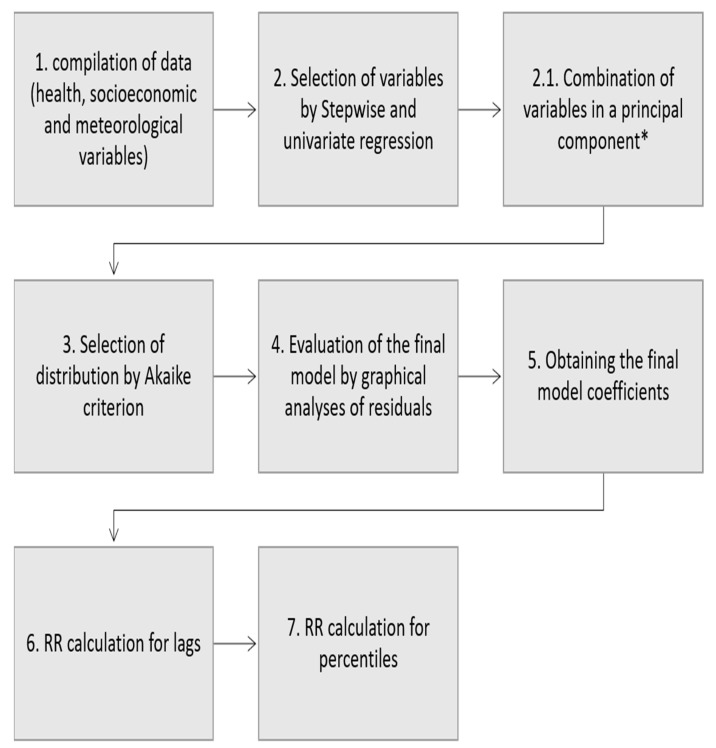
Flowchart of the methodology steps that were performed. * Principal components was applied only in models with correlated variables selected together.

**Figure 3 ijerph-19-16521-f003:**
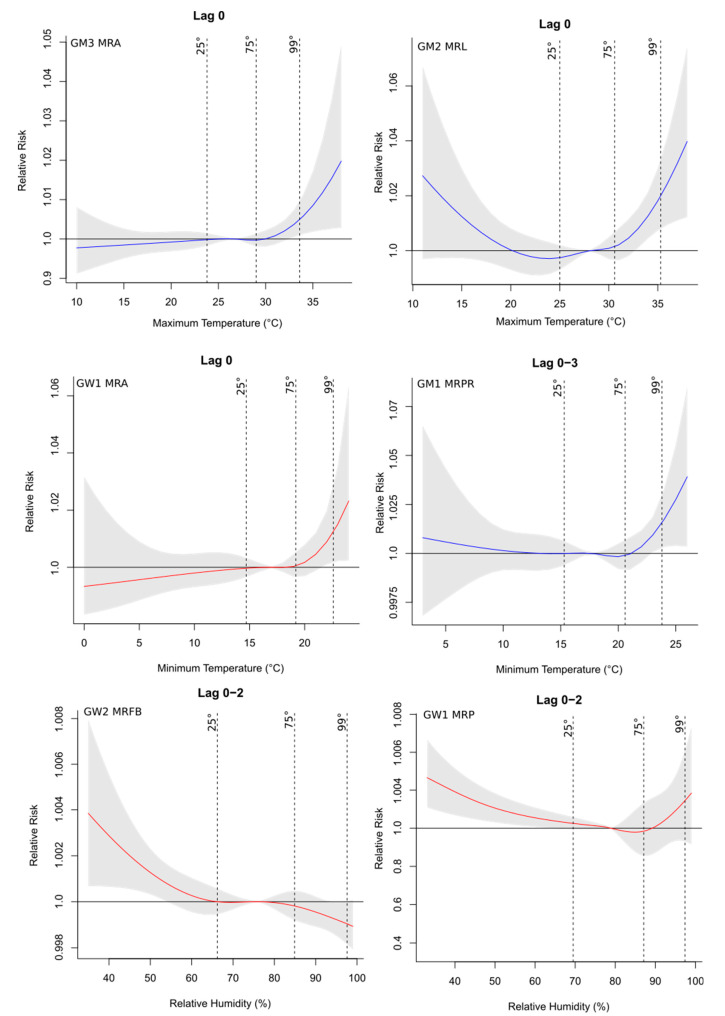
Exposure–response curves of the median variable (reference value according to each region) and cumulative relative risks of hospitalizations (**left** column) and deaths (**right** column) due to cardiovascular diseases for the main lags and groups. Vertical lines indicate the 25th, 75th, and 99th percentiles. The gray region of the plot represents the 95% confidence interval. Blue curves represent groups of men; curves in red represent groups of women.

**Figure 4 ijerph-19-16521-f004:**
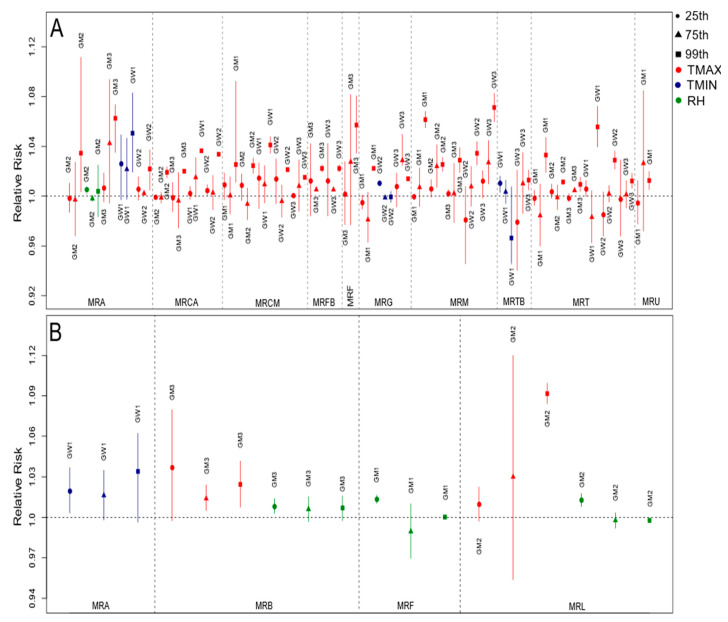
Relative risks of hospitalizations (**A**) and deaths (**B**) due to cardiovascular diseases for groups and regions in the percentile ranges with risk at lag 0. Legend symbols: circles—25th percentile; triangles—75th percentile; squares—99th percentile. Legend color: red—maximum temperature; blue—minimum temperature; green—relative humidity.

**Figure 5 ijerph-19-16521-f005:**
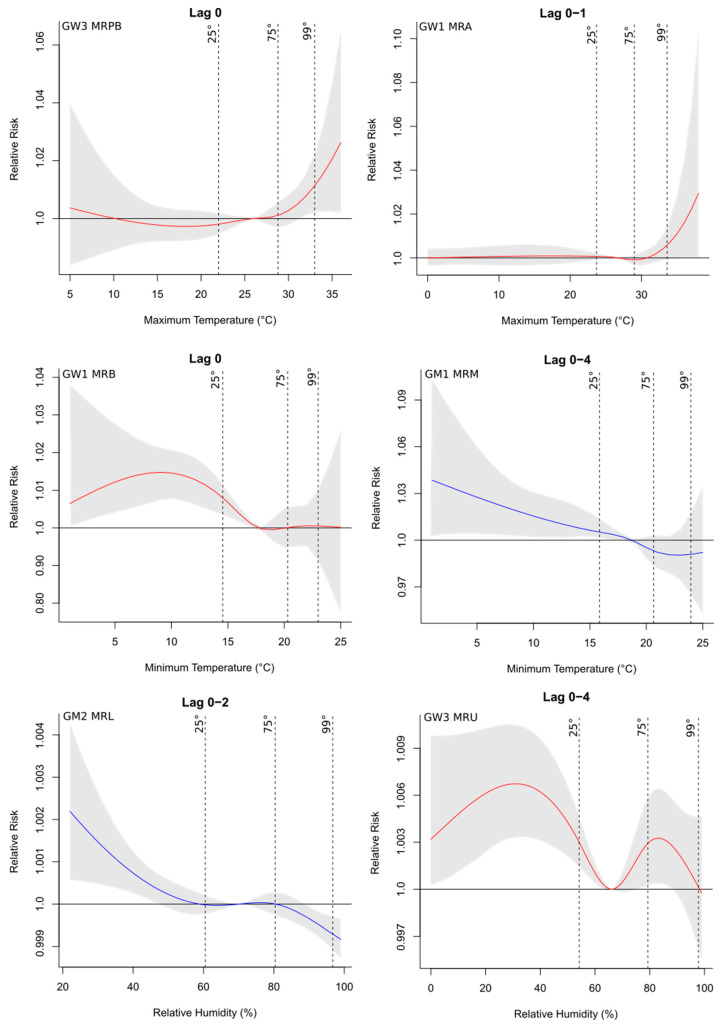
Exposure–response curves of the median variable (reference value according to each region) and the cumulative relative risks of hospitalizations (**left** column) and deaths (**right** column) due to respiratory diseases for the main lags and groups. Vertical lines indicate the 25th, 75th, and 99th percentiles. The gray region of the plot represents the 95% confidence interval. Blue curves represent groups of men; curves in red represent groups of women.

**Figure 6 ijerph-19-16521-f006:**
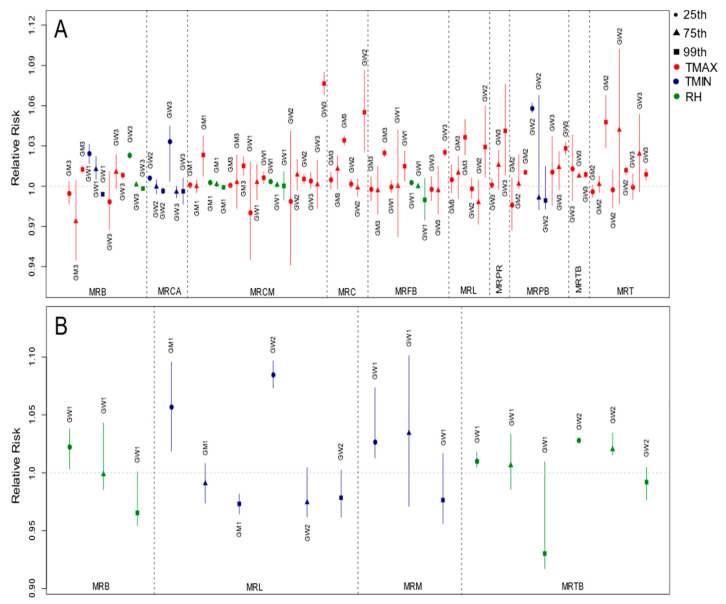
Relative risks of hospitalizations (**A**) and deaths (**B**) due to respiratory diseases for the groups and regions in the percentile ranges with risk at lag 0. Legend symbols: circles—25th percentile; triangles—75th percentile; squares—99th percentile. Legend color: red—maximum temperature; blue—minimum temperature; green—relative humidity.

**Figure 7 ijerph-19-16521-f007:**
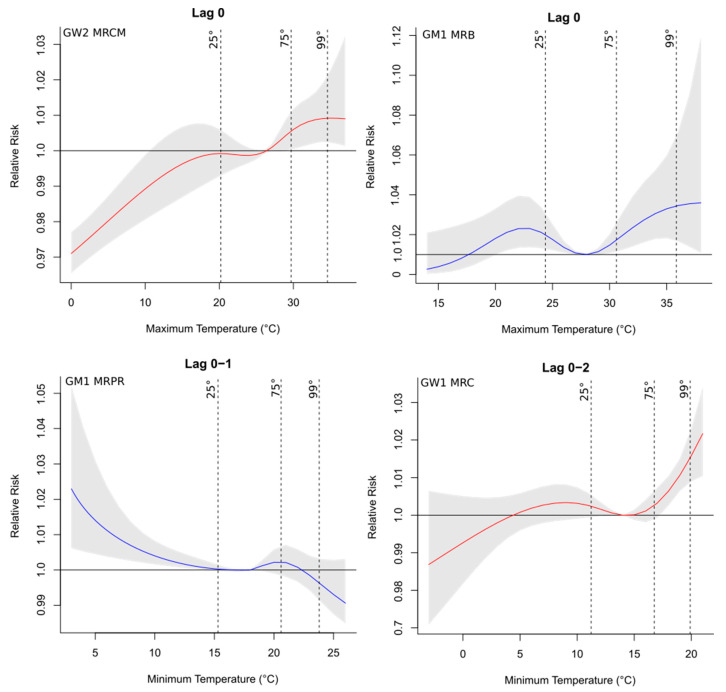
Exposure–response curves of the median variable (reference value according to each region) and the cumulative relative risk of hospitalization for mental illnesses and disorders for the main lags and groups. Vertical lines indicate the 25th, 75th, and 99th percentiles. The gray region of the plot represents the 95% confidence interval. Blue curves represent groups of men; curves in red represent groups of women.

**Figure 8 ijerph-19-16521-f008:**
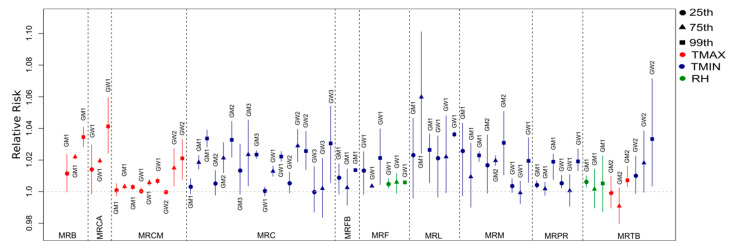
Relative risk of hospitalizations for mental illnesses and disorders for the groups and regions in the percentile ranges with risk at lag 0. Legend symbols: circles—25th percentile; triangles—75th percentile; squares—99th percentile. Legend color: red—maximum temperature; blue—minimum temperature; green—relative humidity.

**Figure 9 ijerph-19-16521-f009:**
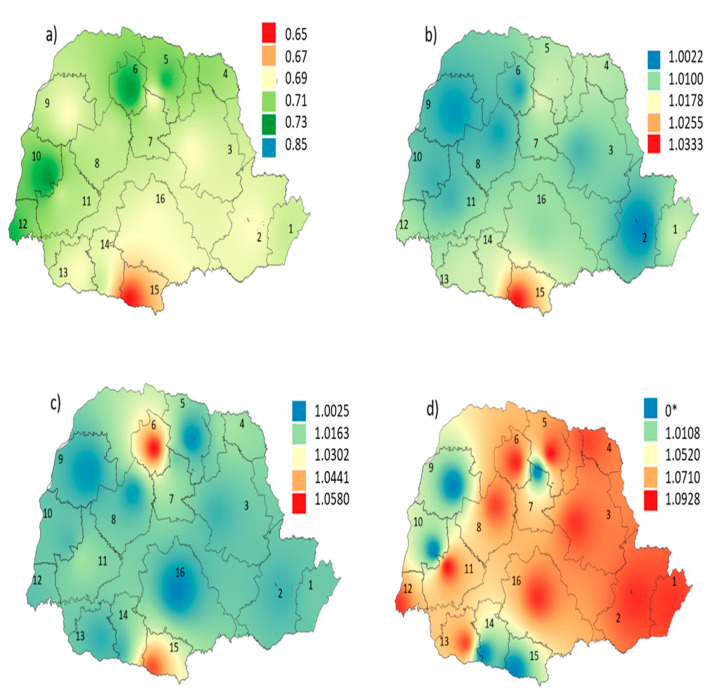
(**a**) Average MHDI for the RM’s (2010 census); Average relative risk of hospitalizations and deaths from: (**b**) cardiovascular diseases; (**c**) respiratory diseases; and (**d**) mental disorders. MRs: Paranaguá (1), Curitiba (2), Telêmaco Borba (3), Bandeirantes (4), Londrina (5), Maringá (6), Apucarana (7), Campo Mourão (8), Umuarama (9), Toledo (10), Cascavel (11), Foz do Iguaçu (12), Francisco Beltrão (13), Pato Branco (14), Palmas (15), and Guarapuava (16). * Exposure–risk relationship not established.

## Data Availability

The datasets analyzed in this study are available online at public institutions: DATASUS (www.datasus.saude.gov.br, accessed on 1 March 2020), IAPAR (www.iapar.br, accessed on 1 March 2020), INMET (www.portal.inmet.gov.br, accessed on 1 March 2020), and SIMEPAR (www.simepar.br, accessed on 1 March 2020).

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
