# Peer review of "Effects of Environmental and Socioeconomic Inequalities on Health Outcomes: A Multi-Region Time-Series Study"

_ijerph, 2022, doi:10.3390/ijerph192416521_

Round 1

Reviewer 1 Report

line 66, 90: why all capitals for in-text citations?

line 160: Abbreviate GM

line 195: Rationale for using binary holiday variables and categorical day of the week variables. Is there an autocorrelation? If yes, present the autocorrelation plot in supplement. 

line 214: are you referring beta estimates as relative risks? Please clarify if they are rate ratios or relative risk estimates.

Methodology is not described clearly. Why selecting certain independent variables is not properly justified. Additionally, please present model diagnostics in supplement. Description of DLNM is not appropriate.

line 235: what is art and mrfb?

Results are confusing as the presentation style is more like discussion than stating the observations.

line 246: abbreviations.

Figure 2. What is the reference temperature or humidity values being used to estimate the relative risks using DLNM. Additionally, describe how you picked the threshold reference value in the algorithm. No figure description provided. No reference number to the individual facets. Few panels titled as lag 0-3, but doesn't show the axis representing lag days.

Figure 3. You're comparing different regions, so I am curious to know if you included population as an offset term in your model. Additionally, figure legend is very confusing, which makes the figure extremely hard to understand. Please use two legends one for color and other for shape.

Reviewer 2 Report

Questions and Suggestions:

1. Calculation method of MDPI should be provided in section 2.3.

2. Line 209-210: meteorological and socioeconomic explanatory variables are x1 is.... , please revise the sentence.

3. I suggest the authors to delete the results of mental disease and focus on the respiratory and cardiovascular diseases, to enhance the results analysis and compare the results between two diseases among different groups under the three percentiles. 

4. To add more information on the results difference between climactic regions in Discussion and give some mechanism explanation if possible. 

Round 2

Reviewer 1 Report

Methods section need detailed explanation on how each model was fit. Figure descriptions need clear explanation for all the figures in the manuscript.

Author Response

Dear reviewer,
thanks for the important suggestions to improve our manuscript.

Other explanations were added in the text regarding the fit of the models, in addition to a flowchart that clarifies the step-by-step of the applied methodology.

Figures descriptions have been improved as requested.

Reviewer 2 Report

no

(Questions 1 and 4 in the last review were not carefully revised. I hope the authors addressed the issues)

Author Response

Dear reviewer,
thanks for the important suggestions to improve our manuscript.

Questions 1 and 4 were addressed, mainly question 1, in this second round. The author believes that it is out of the scope of this work to provide the calculation method of HDMI since it was provided the source and the main dimensions used to calculate the index. The authors use an official index that seeks to capture the socioeconomic conditions of the municipalities in the State of Paraná in its most significant dimensions. The index is calculated and made available by Institute IPARDES, which uses several information to calculate it. It follows a similar line to the Human Development Index (HDI), prepared by the United Nations Development Programme (UNDP). However, the authors prepared a brief description of the methodology and included it in the supplementary material.

 Concerning question 4, the authors already previously included a discussion. The text contains explanations about the mechanism of action of temperature and humidity in the body. However, in this second round, we also added a new paragraph about the relationship between climate and health outcomes. Additional inclusions about what the authors understood to be requested by the reviewer could make the discussion massive and repetitive. Then, to add more information, the authors need more precise indications to fulfill the request.